# PLGA/PEG Nanoparticles Loaded with Cyclodextrin-*Peganum harmala* Alkaloid Complex and Ascorbic Acid with Promising Antimicrobial Activities

**DOI:** 10.3390/pharmaceutics14010142

**Published:** 2022-01-07

**Authors:** Sherif Ashraf Fahmy, Noha Khalil Mahdy, Hadeer Al Mulla, Aliaa Nabil ElMeshad, Marwa Y. Issa, Hassan Mohamed El-Said Azzazy

**Affiliations:** 1Department of Chemistry, School of Sciences & Engineering, The American University in Cairo, AUC Avenue, P.O. Box 74, New Cairo 11835, Egypt; sheriffahmy@aucegypt.edu (S.A.F.); noha.khalil@aucegypt.edu (N.K.M.); hadeer.almulla@aucegypt.edu (H.A.M.); 2Department of Pharmaceutics and Industrial Pharmacy, Faculty of Pharmacy, Cairo University, Kasr El-Aini Street, Cairo 11562, Egypt; aliaa.elmeshad@pharma.cu.edu.eg; 3Department of Pharmaceutics, Faculty of Pharmacy and Drug Technology, The Egyptian Chinese University, Gesr El Suez Street, Cairo 11786, Egypt; 4Department of Pharmacognosy, Faculty of Pharmacy, Cairo University, Kasr El-Aini Street, Cairo 11562, Egypt; marwa.issa@pharma.cu.edu.eg

**Keywords:** *Peganum harmala*, ascorbic acid (Vitamin C), 2-hydroxy propyl-β-cyclodextrin, PLGA, PEG, antiviral, antibacterial

## Abstract

Antimicrobial drugs face numerous challenges, including drug resistance, systemic toxic effects, and poor bioavailability. To date, treatment choices are limited, which warrants the search for novel potent antivirals, including those extracted from natural products. The seeds of *Peganum harmala* L. (*Zygophyllaceae* family) have been reported to have antimicrobial, antifungal, and anticancer activities. In the present study, a 2-hydroxy propyl-β-cyclodextrin (HPβCD)/harmala alkaloid-rich fraction (HARF) host–guest complex was prepared using a thin-film hydration method to improve the water solubility and bioavailability of HARF. The designed complex was then co-encapsulated with ascorbic acid into PLGA nanoparticles coated with polyethylene glycol (HARF–HPßCD/AA@PLGA-PEG NPs) using the W/O/W multiple emulsion-solvent evaporation method. The average particle size, PDI, and zeta potential were 207.90 ± 2.60 nm, 0.17 ± 0.01, and 31.6 ± 0.20 mV, respectively. The entrapment efficiency for HARF was 81.60 ± 1.20% and for ascorbic acid was 88 ± 2.20%. HARF–HPßCD/AA@PLGA-PEG NPs had the highest antibacterial activity against *Staphylococcus aureus* and *Escherichia coli* (MIC of 0.025 mg/mL). They also exhibited high selective antiviral activity against the H1N1 influenza virus (IC50 2.7 μg/mL) without affecting the host (MDCK cells). In conclusion, the co-encapsulation of HPCD–HARF complex and ascorbic acid into PLGA-PEG nanoparticles significantly increased the selective H1N1 killing activity with minimum host toxic effects.

## 1. Introduction

Microbial infections, including bacterial and viral infections, represent a significant health burden responsible for many deaths worldwide. Influenza A virus causes the hospitalization of about 4 million cases and the death of 600,000 patients each year. The pervasive use of antimicrobials is the leading cause of resistance to the existing medications [1]. Moreover, most antimicrobial agents suffer from many drawbacks, including systemic adverse reactions, allergy, poor bioavailability, and narrow spectrum [2,3].

Nature provides an enormous library of novel chemicals that can be explored to develop antimicrobials. *Peganum harmala* L. (*Zygophyllaceae* family) is a member of the *Zygophyllaceae* family native to the area from the eastern Mediterranean region to India [3,4]. The seeds of *P. harmala* are rich in harmala alkaloids such as quinazoline and β-carboline alkaloids. The major alkaloid, harmine, is responsible for the various pharmacological activities of the harmala seeds [5,6,7]. Several studies have reported the promising antimicrobial activities of *P. harmala* extracts [8,9].

Ascorbic acid is an essential dietary nutrient required for various biological functions [10]. Several in vivo studies have reported the ability of ascorbic acid to prevent and alleviate many types of viral infections [10,11]. Ascorbic acid was found to improve the immune response to viral infections by stimulating the function and proliferation of T-lymphocyte and NK-lymphocyte and the production of interferon [11].

Despite the global trend to replace synthetic antimicrobials with natural ones, the use of natural products to treat microbial infections may face challenges similar to those of synthetic drugs, such as poor water solubility, low bioavailability, and non-selective targeting the infected organ. Thus, numerous nanoplatforms, including liposomes, supramolecular systems, and polymeric nanoparticles, were designed, being reported to remarkably ameliorate the therapeutic activities of different types of biologically active compounds [12,13,14,15,16,17,18].

Supramolecular host molecules, such as cyclodextrins (CDs), are involved in overcoming the poor water solubility of the natural compounds via the formation of host–guest complexes. CDs are amphiphilic cyclic oligosaccharides obtained from the enzymatic hydrolysis of starch. They comprise three types, namely, α, β, and γ-CDs, consisting of six, seven, and eight glucopyranose units, respectively [19]. In this context, 2-hydroxy propyl ß cyclodextrin (HPßCD) is a promising derivative of CDs due to its safety, high water solubility, and ability to selectively form inclusion complexes with various drug molecules [20].

Polymeric nanoparticles (NPs) have been extensively involved in drug delivery, owing to their ability to encapsulate different therapeutically active compounds into their matrix and the ability of their surfaces to be decorated with different functional groups [3,21,22]. Consequently, this could improve the stability of their cargos and prolong their half-lives. Poly (lactic-co-glycolic acid) (PLGA) is commonly employed in the fabrication of polymeric NPs since it is a biocompatible and biodegradable polymer with favorable physicochemical properties [3,21,22]. The surface decoration of PLGA NPs with polyethylene glycol (PEG) has been reported to have many advantages, such as (i) improving the water solubility of the PLGA NPs and their hydrophobic cargos, (ii) increasing the surface charge and the steric hindrance on the PLGA NPs thus extending the stability of the NPs and minimizing their aggregation, (iii) causing a stealth effect that prolongs the systemic circulation duration of drugs via inhibiting the opsonization effect, and (iv) enhancing the release of the drugs out of the polymeric matrix [23,24]. Many studies have reported using PEGylated PLGA NPs as promising nanocarriers for various natural bioactive compounds, biotherapeutics (as peptides, proteins, or vaccines), and synthetic drugs [23,24,25,26,27].

In the present study, a 2-hydroxy propyl-β- cyclodextrin (HPβCD)/harmala alkaloid-rich fraction (HARF) host–guest complex was fabricated using the thin-film hydration method to alleviate the water solubility of HARF. The complex formation was investigated by ^1^H NMR spectroscopy and phase solubility study. The prepared complex was then co-encapsulated with ascorbic acid into PEGylated PLGA (HARF–HPßCD/AA@PLGA-PEG NPs) utilizing the W/O/W multiple emulsion-solvent evaporation method. The designed NPs were physically characterized in terms of average size, surface charge, polydispersity index (PDI), entrapment efficiency, and morphology. Additionally, the release of HARF–HPβCD complex and ascorbic acid (AA) from HARF–HPßCD/AA@PLGA-PEG NPs was studied, and the release profile was acquired. Finally, the antibacterial and antiviral activities of HARF–HPßCD/AA@PLGA-PEG NPs were evaluated.

## 2. Materials and Methods

### 2.1. Materials

2-Hydroxy propyl ß cyclodextrin was purchased from BLD Pharmatech Co., Limited (Cincinnati, OH, USA). Poly (D,L-lactide-co-glycolide) (PLGA) and polyethylene glycol 6000 (PEG) were obtained from Sigma-Aldrich (St. Louis, MO, USA). Polyvinyl alcohol (PVA; 98% hydrolyzed, MW ≈ 13,000) was obtained from Acros Organics (Geel, Belgium). Dimethylsulfoxide was purchased from Fisher Chemicals (Fair Lawn, NJ, USA). Tween 80 was purchased from El-Nasr Pharmaceutical Chemicals Co. (Cairo, Egypt). Tryptic soy agar was purchased from Millipore (Bedford, MA, USA). Phosphate-buffered saline, streptomycin, penicillin, fetal bovine serum, trichloroacetic acid, Dulbecco’s modified Eagle’s medium (DMEM), SRB, and tris (hydroxymethyl)aminomethane were purchased from Lonza (Basel, Switzerland).

### 2.2. Extraction and Isolation of Major P. harmala Alkaloids

Dried mature seeds of *P.*
*harmala* L. were purchased from the local Egyptian market. The major alkaloids of *P. harmala* seeds were extracted, isolated, and characterized according to our previously reported method [5,6].

### 2.3. Preparation of Harmala Alkaloid-Rich Fraction/2-Hydroxy Propyl ß Cyclodextrin Complex (HARF–HPßCD)

HARF–HPßCD inclusion complex was designed via the thin-film hydration method, as described elsewhere, with some modifications [28]. Briefly, a thin film was formed by dissolving HARF in methanol followed by drying in a round bottom flask using a rotary evaporator. Separately, 5% (*w*/*v*) of HPßCD solution was prepared and subsequently added to the film, followed by 30 min sonication in a SONOREX DIGITAL 10 P sonicator (BANDELIN electronic GmbH & Co. KG, Darmstadt, Germany). Afterward, the complex solution was stirred for another 30 min and purified through filtration using a 0.22 μm nylon filter. The filtrate was then dried in a lyophilizer (TOPTION TOPT-10C Freeze dryer, Toption Group Co., Xi’an, China).

#### 2.3.1. ^1^H NMR Spectroscopy of HARF–HPßCD Complex

^1^H NMR spectra of HPßCD and the inclusion complex were measured on a 400 MHz FT-NMR spectrometer (ECA-500, JEOL, Tokyo, Japan) in D_2_O solutions. The scanning range of the hydrogen spectrum is 1–13 ppm.

#### 2.3.2. Phase Solubility Study

The phase solubility study was performed, as described elsewhere with some modifications [29], to confirm the formation of the HARF–HPßCD inclusion complex using UV spectroscopy (FLUOstar Omega microplate reader, BMG Labtech, Offenburg, Germany) at λ = 373 nm [30]. Briefly, an excess amount of HARF was added to 5 mL of several mixtures containing successively increasing concentrations of HPßCD (ranging from 0 to 200 mm) all in deionized water. These mixtures were then sealed and stirred at room temperature until equilibrium had been reached. The mixtures were then centrifuged at 4000 rpm for 10 min (Hermle Z326K, Wehingen, Germany), and the absorbance was measured at 373 nm using UV spectrophotometer.

The stability constant (*K*_1:1_) was estimated utilizing the slope of the linear correlation and the intrinsic solubility (*S*_0_) using Equation (1) [31,32].
(1)$$K_{1:1}=\frac{Slope}{S_{0}\left(1-Slope\right)}$$

Then, the complexation efficiency (*CE*) was computed employing the intrinsic solubility and the stability constant using Equation (2) [31,32].
(2)$$CE=S_{0}*\;K_{1:1}$$
where *S*_0_ is the intrinsic solubility, and *K*_1:1_ is the stability constant.

### 2.4. Preparation of PEG-Coated PLGA Nanoparticles Dual-Loaded with Ascorbic Acid (AA) and HARF–HPßCD Inclusion Complex (HARF–HPßCD/AA@PLGA-PEG NPs)

The designed HARF–HPßCD complex was co-loaded with AA into PLGA NPs coated with PEG (HARF–HPßCD/AA@PLGA-PEG NPs) using the W/O/W multiple emulsion-solvent evaporation method, as previously reported with some modifications [3,7,33,34].

Briefly, 300 mg of PLGA and 100 mg PEG 6000 were dissolved in 8 mL dichloromethane (DCM), forming the emulsion’s organic phase (O). Then, HARF–HPßCD complex and AA were dissolved in water (the W/W ratio of each drug/PLGA was 0.2:1) to obtain the inner aqueous phase (W1). Into the previously prepared organic phase, we emulsified 800 µL of the internal aqueous phase using a homogenizer at 11,000 rpm for 8 min in an ice bath to form a W1/O emulsion (first emulsion). Then, 2.5% polyvinyl alcohol was dissolved in 100 mL deionized water to form the external aqueous phase (W2). The first emulsion was added to the external aqueous phase (W2) and emulsified by homogenization at 11,000 rpm for 8 min in an ice bath. The resulting W1/O/W2 emulsion (second emulsion) was stirred overnight using a magnetic stirrer to allow solvent evaporation and particle hardening. Afterward, the nanoparticle colloid was lyophilized (TOPTION TOPT-10C Freeze dryer, Toption Group Co., Limited).

### 2.5. Characterization of the Designed HARF–HPßCD/AA@PLGA-PEG NPs

The average particle size and polydispersity index (PDI) were determined using a Zetasizer Nano ZS equipped with a 10 mW HeNe laser, allowing for measurements at 633 nm and a detection angle of 173° backscatter (Malvern Instruments, Worcestershire, UK). The ζ-potential was measured using laser Doppler velocimetry [35,36,37].

The morphological features of the prepared nanoparticles were investigated using scanning electron microscopy (SEM) via high-resolution field emission scanning electron microscopy (FESEM) (LEO SUPRA 55, Carl Zeiss, Oberkochen, Germany) with an acceleration voltage of 1.00 kV. Before conducting the SEM study, we coated freeze-dried PLGA NPs via gold sputtering (current: 10 mA) for 2 min under a nitrogen atmosphere. The measurements were performed using the image processing program ImageJ (NIH, Bethesda, MD, USA).

### 2.6. Entrapment Efficiency

The entrapment efficiency (*EE*) of HARF–HPßCD/AA@PLGA-PEG NPs was measured indirectly by centrifugation (15,000 rpm, 1 h, 4 °C), followed by ultrafiltration of the supernatant. The quantities of the HARF and AA unencapsulated in the ultrafiltrate were determined by a FLUOstar Omega microplate reader (BMG Labtech, Offenburg, Germany) spectrophotometer at 373 and 287 nm, respectively, using Equation (3) [37,38,39].
(3)$$EE\;\left(\%\right)=\frac{Initial\;feeded\;drug-Final\;loaded\;drug}{Initial\;feeded\;drug}\;\times \;100$$

### 2.7. In Vitro Release Study of HARF–HPßCD Complex and Ascorbic Acid from HARF–HPßCD/AA@PLGA-PEG NPs

The cumulative release % of HARF and AA from HARF–HPßCD/AA@PLGA-PEG NPs was investigated utilizing the dialysis membrane method. Briefly, 1 mL of the NPs was loaded to a dialysis bag (cutoff molecular weight of 12,000–14,000 Da). The dialysis bag was inserted into 50 mL of PBS at pH 7.4 with 2% Tween in a proper jar. The whole system was left on a stirrer at 37 °C. A 1 mL aliquot of the sample was withdrawn for analysis and immediately replaced with another equal volume of warmed buffer at specific time intervals. The concentrations of HARF and AA were determined by a FLUOstar Omega microplate reader (BMG Labtech, Offenburg, Germany) spectrophotometer at 373 and 287 nm wavelengths, respectively.

The cumulative release (*%*) was calculated according to Equation (4) [40].
(4)$$Cumulative\;release\;\left(\%\right)=\frac{\mathrm{Vs}\sum_{1}^{n-1}c\left(n-1\right)+VoCn}{\mathrm{mo}}\times 100$$
where Vs is the volume of each sample removed, *c*(*n* − 1) is the bulk concentration before sampling, *Cn* is the concentration of the sample, *Vo* is the bulk volume of the release medium, and mo is the original amount of the drug in the nanoparticles tested.

### 2.8. Antibacterial Assay for the Designed HARF–HPßCD/AA@PLGA-PEG NPs

#### 2.8.1. Preparation of the Inoculum Using Colony Suspension Approach

*Staphylococcus aureus* ATCC^®^ 6538 (lot no. 4600502) and *Escherichia coli* ATCC^®^ 8739 (lot no. 380063) were obtained from American Type Culture Collection (Manassas, VA, USA). Tryptic soy broth (100 mL) was used as inoculation medium for either the Gram-positive or Gram-negative bacteria, then incubated at 37.0 ± 1.0 °C for 24 ± 2 h. A loopful of the broth was streaked onto tryptic soy agar medium and incubated at 37.0 °C for 21 ± 3 h. Three or four colonies (from each plate) were inoculated in broth, and the suspension was incubated till turbidity reached 0.5 McFarland standard. The inoculum density was standardized utilizing a 0.5 McFarland standard and DensiCHEK^©^ optical device (BioMérieux, Marcy l’Etoile, France). The obtained suspensions contained about 1.0 × 10^8^ CFU/mL of *Staphylococcus aureus* or *Escherichia coli*.

#### 2.8.2. Broth Macrodilution Method

The antibacterial activities of plain PLGA-PEG NPs, HARF, and HARF–HPßCD/AA@PLGA-PEG NPs were evaluated by adding and mixing 5.0 mL from each sample to 5.0 mL broth (1:2 dilution) according to a previously reported method [41]. Next, 5.0 mL of the 1:2 dilution was pipetted using a new tip and mixed with 5.0 mL broth (1:4). Ten dilutions for each bacterial strain were prepared and inoculated in a 24-well plate. Then, 100 µL of prepared inoculum was added to each well, resulting in a final concentration of 5.0 × 10^5^ CFU/mL. An additional 100 µL from each bacterial strain suspension was diluted and cultured to confirm the inoculum density. All plates were incubated at 37.0 °C for 24 h and then kept in the dark to monitor growth. All control wells yielded a turbid solution. The inoculum density was 4–6 × 10^5^ CFU/mL for both evaluated bacterial strains, with reference to the Clinical and Laboratory Standards Institute (CLSI) procedures (document M07, A09) [41].

### 2.9. Cytotoxicity and Antiviral Assays

The human influenza H1N1 and Madin–Darby Canine Kidney (MDCK) cells were obtained from American Type Culture Collection (University Boulevard, Manassas, VA, USA). Vero E6 cells were grown in DMEM medium supplemented with 10% fetal bovine serum and 0.1% antibiotic/antimycotic solution. Gibco BRL provided the antibiotic and antimycotic solution, trypsin-EDTA, fetal bovine serum, and DMEM medium (Grand Island, NY, USA).

Cytopathic effect (CPE) reduction assay was used to evaluate the antiviral activities of plain PLGA-PEG NPs, HARF, and HARF–HPßCD/AA@PLGA-PEG NPs in cell culture systems, as described elsewhere with some modifications [42,43]. Briefly, MDCK cells seeded into a 96-well culture plate at a density of 2 × 10^4^ cells per well were infected with 0.1 mL (CCID50) influenza (H1N1) of the diluted viral suspension of Influenza H1N1 and then incubated at 37 °C for 60 min to facilitate the adsorption of the virus. The antiviral activity for each sample was determined using a twofold diluted concentration of 1–100 µg/mL. Culture plates were incubated at 37 ^°^C in 5% CO_2_ for 72 h. The development of the cytopathic effect was monitored by light microscopy. The cell monolayers were stained (0.03% crystal violet in 2% ethanol and 10% formalin). After washing and drying, the optical density was measured at 540/630 nm. The antiviral activities were calculated according to Pauwels et al. [44], employing Equation (5).
(5)$$Antiviral\;activity\;\%=\frac{\left(\mathrm{mean}\;\mathrm{optical}\;\mathrm{density}\;\mathrm{of}\;\mathrm{cell}\;\mathrm{controls}-\mathrm{mean}\;\mathrm{optical}\;\mathrm{density}\;\mathrm{of}\;\mathrm{virus}\;\mathrm{controls}\right)}{\left(\mathrm{Optical}\;\mathrm{density}\;\mathrm{of}\;\mathrm{the}\;\mathrm{test}-\mathrm{mean}\;\mathrm{optical}\;\mathrm{density}\;\mathrm{of}\;\mathrm{virus}\;\mathrm{controls}\right)}\times 100$$

The cytotoxicity was evaluated against MDCK cells before conducting the aforementioned assay. Cells were seeded at a density of 2 × 10^4^ cells per well in a 96-well culture plate. After 24 h, the culture medium containing serially diluted samples was added to the cells and incubated for 72 h before being removed, and the cells were washed with PBS. The next steps were carried out in the same manner as described above for the antiviral activity assay. The 50% cytotoxic concentrations (CC50) and the 50% inhibitory concentration (IC50) were computed using GraphPad PRISM Software (Graph-Pad Software, San Diego, CA, USA). Selectivity index (SI) was estimated as the ratio of CC50 to IC50.

## 3. Results and Discussion

### 3.1. Characterization of the Prepared HARF–HPßCD Complex

#### 3.1.1. ^1^H NMR Spectroscopy of the HARF–HPßCD Complex

^1^H NMR analyses performed in D_2_O could shed more light on the host–guest complexation between HPßCD and other guest molecules and could provide clues on the inclusion of guest molecules into the hydrophobic cavity of the host molecule. Upon the inclusion of the guest molecule into the HPßCD cavity, an upfield shift of the H-3 and H-5 (hydrogen atoms placed in the interior cavity of HPßCD) occurred due to their shielding by the guest molecule (the chemical structure of HPßCD is presented in Appendix A). On the other hand, H-1, H-2, H-4, and H-6 (hydrogen atoms localized on the outer surface of HPßCD) will experience milder chemical shifts. In the present work, the suggestion of the HARF–HPßCD complex foundation was grounded on modifying the ^1^H NMR spectra of the pure HPßCD upon its complexation with HARF (Appendix A). The ^1^H NMR chemical shifts (in ppm) of HPßCD protons in the presence and absence of HARF are presented in Table 1. The induced chemical shift, δ (ppm), is the difference in chemical shifts of HPßCD protons before and after the formation of the HARF–HPßCD complex. In this context, a positive sign of δ (ppm) indicates a downfield shift, while a negative sign indicates an upfield one. It was shown that the H-3 and H-5 protons experienced more remarkable HARF-induced chemical shifts than the outer surface protons (H-1, H-2, H-4, and H-6), indicating that HARF was only included within the HPßCD cavity. Furthermore, the clear upfield shift of H-3 and H-5 protons signals was due to the inclusion of π-electron-rich aromatic groups of HARF inside the host cavity. These findings agree very well with the previous studies, which reported that the upfield proton shifts of H-3 and H-5 indicate inclusion complexation [45,46,47,48].

#### 3.1.2. Phase Solubility Study

The solubility of HARF was determined by plotting the change with HP*β*CD concentration, as presented in Figure 1. The *R*^2^ value of the extrapolated curve was 0.9807, suggesting that a remarkable linear correlation between HARF solubility and HP*β*CD concentration where the solubility of HARF increased with increasing the concentration of HP*β*CD. As previously described by Higuchi and Connors, this phase solubility diagram displays the A_L_-type pattern, indicating a complex stoichiometry of 1:1 (HARF/HP*β*CD) [29,31]. Moreover, the slope was >0.0 and <1.0, while the K value was >0.0, confirming the formation of a 1:1 complex [31].

The stability constant and the complexation efficiency (CE) were 81.4 M^−1^, and 0.66, respectively, similar to values reported previously [31,32,49]. This showed that the HP*β*CD had improved the water solubility of HARF via the formation of the host–guest complex.

### 3.2. Characterization of the HARF–HPßCD/AA@PLGA-PEG NPs

#### 3.2.1. Particle Size, Polydispersity Index (PDI), ζ-Potential, Entrapment Efficiency (EE), and Morphological Features

The average particle size and PDI of the designed HARF–HPßCD/AA@PLGA-PEG NPs were 207 ± 2.60 nm and 0.17 ± 0.01, respectively (Table 2). These values lie within the range of the previously reported nanoparticles that exhibited improved biomedical applications [50,51]. The particle size distribution of the nanoparticles is presented in Figure 2. The ζ-potential of the designed nanoparticles showed a high negative surface charge of −31.60 ± 0.20 mV. This is attributed to the presence of the highly negatively charged PEG coating the PLGA NPs. Using PEG as a coat for the NPs improves the stability of the PLGA NPs in aqueous media, enabling their long shelf life. Moreover, PEG inhibits the aggregation of NPs through its steric hindrance effect [49].

The entrapment efficiencies of the HARF and AA are presented in Table 2.

The high entrapment efficiencies of HARF and AA are attributed to the PEG coating the PLGA NPs founding a barrier that surrounds and protects the loaded drugs, preventing their loss during the homogenization step and retaining their stability [52]. The capacity to entrap high concentrations of drugs supports sustained drug release [53].

As demonstrated in Figure 3A, SEM analysis showed a spherical shape of the prepared NPs. The mean average size of the NPs was determined by utilizing the image processing program ImageJ (NIH, Bethesda, MD, USA) and was found to be 204.5 ± 70.6 nm (Figure 3B, which is close to that obtained from the dynamic light scattering (207 ± 2.6).

#### 3.2.2. In Vitro Release Study

Figure 4 portrays the release profile of HARF–HPßCD complex and AA from the HARF–HPßCD/AA@PLGA-PEG NPs. The release profile showed an initial release of 29.8 ± 4.9% and 66.7 ± 6.8% for HARF–HPßCD complex and AA, respectively, in the first 2 h, which increased to 64.6 ± 3% and 76.6 ± 1.3%, respectively, over 48 h. These release profiles are attributed to the hydrolytic cleavage of the ester bonds of the PLGA copolymer in the presence of water [54]. Moreover, coating NPs with PEG was reported to enhance the permeability and release of the loaded drugs [55]. The faster release rates of AA compared to HARF complex could be due to the higher molecular weight of the complex, making its diffusion out of the dialysis membrane much slower than that of AA.

#### 3.2.3. Antibacterial Assay for the Designed HARF–HPßCD/AA@PLGA-PEG NPs

The bacterial activities of the HARF, plain PLGA-PEG NPs, and HARF–HPßCD/AA@PLGA-PEG NPs were assessed against *S. aureus* and *E. coli*. HARF–HPßCD/AA@PLGA-PEG NPs exhibited high antibacterial activity against *S. aureus* and *E. coli* (MIC of 0.025 mg/mL), as compared to free HARF (MIC of 0.5 mg/mL), as shown in Table 3. On the other hand, PLGA-PEG NPs showed no antibacterial activities. The HARF–HPßCD/AA@PLGA-PEG NPs showed higher bactericidal activity as compared to free HARF. This could possibly be attributed to the improved hydrophilicity of HARF upon its inclusion inside the HPßCD cavities. Furthermore, the NPs can adhere to the bacterial cell wall due to their effective surface areas, releasing their cargo across the cell wall [34,56,57,58]. The bactericidal activity of HARF is attributed to the presence of high concentrations of β-carboline and quinazoline alkaloids, which have been reported to intercalate the bacterial DNA [6].

#### 3.2.4. Cytotoxicity and Antiviral Activity Assays

Treatment choices for viral infections are limited and face many challenges, such as drug resistance, systemic toxic effects, and poor bioavailability. This warrants the search for effective antiviral agents that can overcome the shortcomings of their current counterparts. The antiviral activities of the HARF, PLGA-PEG NPs, and HARF–HPßCD/AA@PLGA-PEG NPs against influenza A (H1N) were investigated through employing the cytopathic effect (CPE) reduction assay. The PLGA-PEG showed no antiviral activity. The developed HARF–HPCD/PLGA-PEG NPs exhibited high selective antiviral activity against the H1N1 influenza virus (Table 4) without affecting the host, MDCK cells, (IC50 of 2.7 μg/mL; CC50 of 110.4 μg/mL; and selective index, CC50/IC50, of 41.2) compared to free HARF (IC50 of 30.2 μg/mL, CC50 of 238.8 μg/mL, and selective index of 7.9). The NPs decompose, releasing their cargos that kill the virus selectively without harming the host cells. HARF is reported to exert its antiviral activity by inhibiting the viral polymerase activity and consequently viral RNA replication [59]. On the other hand, the co-encapsulation of ascorbic acid causes a synergistic antiviral effect against the influenza virus by increasing the generation of interferon (IFN)-α/β [60,61].

## 4. Conclusions

In this work, HARF was first mixed with HPßCD as a host molecule to form an inclusion complex. This was done to improve the solubility of HARF, which is highly hydrophobic. Finally, AA and HARF–HPßCD inclusion complex were co-loaded into the PLGA-PEG nanoparticles to form the final biodegradable delivery system with enhanced release and bioavailability. The fabricated HARF–HPßCD/AA@PLGA-PEG NPs demonstrated acceptable cytotoxicity and enhanced antibacterial and antiviral activities compared to free HARF. They showed a higher bactericidal effect against *Staphylococcus aureus* and *Escherichia coli*, with a MIC value of 0.025 mg/mL, as compared to free HARF (MIC of 0.50 mg/mL). Moreover, the designed NPs exhibited high selective antiviral activity against the H1N1 influenza virus without affecting the host, MDCK cells. Biocompatible nanocarriers loaded with natural extracts with antimicrobial activity could be explored as future effective treatments of infections.

## Figures and Tables

**Figure 1 pharmaceutics-14-00142-f001:**
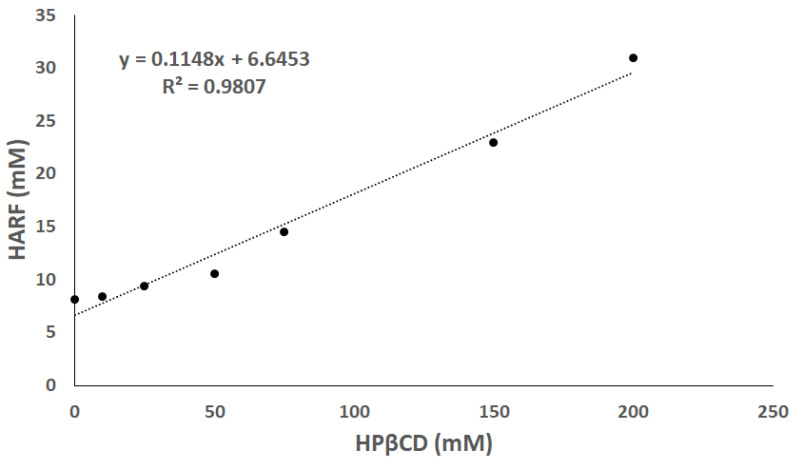
Phase solubility diagram of HARF in the presence of several mixtures containing successively increasing concentrations of HP*β*CD.

**Figure 2 pharmaceutics-14-00142-f002:**
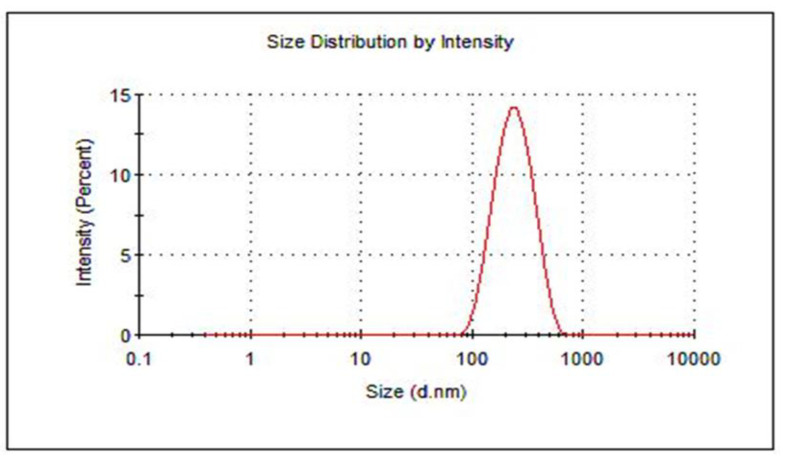
Particle size distribution of the prepared HARF–HPßCD/AA@PLGA-PEG NPs using DLS analysis.

**Figure 3 pharmaceutics-14-00142-f003:**
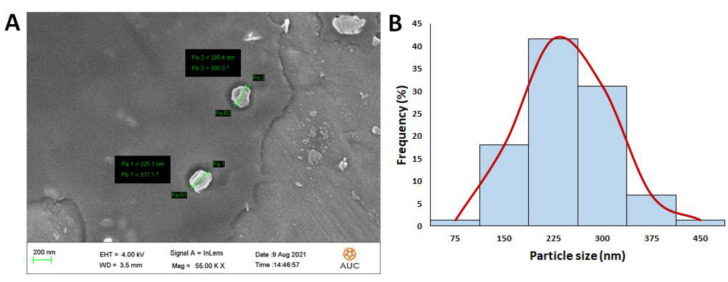
(**A**) Scanning electron microscopy (SEM) images of the HARF–HPßCD/AA@PLGA-PEG NPs. (**B**) Particle size (nm) histogram of the designed HARF–HPßCD/AA@PLGA-PEG NPs generated through employing the image processing program ImageJ (NIH, Bethesda, MD, USA).

**Figure 4 pharmaceutics-14-00142-f004:**
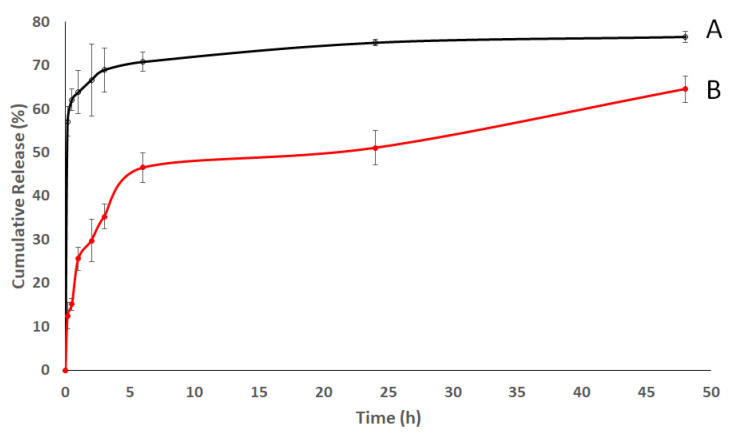
Time-dependent release profiles of (**A**) AA and (**B**) HARF–HPßCD complex from HARF–HPßCD/AA@PLGA-PEG NPs at 37 °C into phosphate-buffered saline.

**Table 1 pharmaceutics-14-00142-t001:** The chemical shifts (ppm) of HP*β*CD before and after forming the HARF–HPßCD complex.

	Chemical Shifts (δ, ppm)					
	H-1	H-2	H-3	H-4	H-5	H-6
HPßCD	4.899	3.310	3.768	3.329	3.552	3.696
HARF-HPßCD complex	4.877	3.307	3.741	3.307	3.526	3.675
Δδ	−0.022	−0.003	−0.027	−0.022	−0.026	−0.021

**Table 2 pharmaceutics-14-00142-t002:** Average particle size, PDI, ζ-potential, and entrapment efficiency of HARF–HPßCD/AA@PLGA-PEG NPs.

Formula	Average Size (nm)	PDI	ζ-Potential (mV) ± SD	Encapsulation Efficiency (%)	
				HARF	AA
HARF–HPßCD/AA@ PLGA-PEG NPs	207 ± 2.60	0.17 ± 0.01	−31.60 ± 0.20	81.60 ± 1.20	87 ± 2.20

**Table 3 pharmaceutics-14-00142-t003:** Bactericidal activity of HARF and HARF–HPßCD/AA@PLGA-PEG NPs against *Staphylococcus aureus* and *Escherichia coli*.

Bacterial Strain	Minimum Inhibitory Concentration (MIC in mg/mL)		
	PLGA-PEG NPs	HARF	HARF–HPßCD/AA@PLGA-PEG NPs
*Staphylococcus aureus*	0	0.5	0.025
*Escherichia coli*	0	0.5	0.025

**Table 4 pharmaceutics-14-00142-t004:** Antiviral activity of HARF and HARF–HPßCD/AA@PLGA-PEG NPs against influenza A (H1N1) virus.

Sample	CC50 (μg /mL)	IC50 (μg /mL)	SI
Plain PLGA-PEG	0	0	0
HARF	238.8	30.2	7.9
HARF–HPßCD/AA@PLGA-PEG NPs	110.4	2.7	41.2

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
