# Peer review of "PLGA/PEG Nanoparticles Loaded with Cyclodextrin-Peganum harmala Alkaloid Complex and Ascorbic Acid with Promising Antimicrobial Activities"

_pharmaceutics, 2022, doi:10.3390/pharmaceutics14010142_

Round 1

Reviewer 1 Report

The manuscripts by Fahmy et al described the effect of antibacterial and antiviral activities of HARF-HPßCD/AA@PLGA-PEG NPs. The manuscript is well written and organized. However, authors must revise the several minor problems.

  1. Line 158, what is the X?It would be helpful for the reader, if the authors add the information for each variable of the equation.
  2. Line 194, 225, 227, 228, it would be better if the authors re-organize the notation, ex: oC, 3 – 4, etc.
  3. Line 236 “afterwards, 5.0 mL was pipetted by new tip”. 5 mL of what solution? It would be better if the authors clarify the that sentence.
  4. Line 237, why did the authors perform ten-time dilutions in the experiment?
  5. Line 241, why did the authors choose 5 x 105 CFU/mL as a final concentration ?
  6. Line 243, “ was added to each organism/sample” The underline can be removed

Reviewer 2 Report

Dear Editor, Dear Authors,

I evaluated the manuscript « PLGA/PEG Nanoparticles loaded with Cyclodextrin- Peganum harmala alkaloid complex and Ascorbic Acid with Promising Antimicrobial Activities » by Sherif Ashraf Fahmy et al.

In their study, the authors investigated antimicrobial activities of a 2-hydroxy propyl-β- cyclodextrin (HPβCD)/harmala alkaloid-rich fraction (HARF) co-encapsulated with ascorbic acid into PLGA nanoparticles coated with polyethylene glycol (HARF40 HPßCD/AA@PLGA-PEG NPs). Authors found that the prepared NPs exerted an increased antibacterial  activity against Staphylococcus aureus and Escherichia coli (MIC of 0.025 mg/mL) compared to HARF (MIC of 0.50 mg/mL). In addition, data demonstrate that the designed HARF-HPßCD/AA@PLGA-PEG NPs exhibited high selective antiviral activity against the H1N1 influenza virus without toxicity against human cells (IC50 2.7 mg/mL, CC50 110.4 mg/mL, and selective index, CC50/IC50, 41.2) compared to free HARF (IC50 30.2 mg /mL, CC50 238.8 mg/mL, and selective index 7.9). The conclusions of the authors is that the co-encapsulation of HPCD-HARF complex and ascorbic acid into PLGA-PEG nanoparticles significantly increases its antimicrobial and antiviral activity with decreasing its toxicity.

I found the manuscript interesting and the study well conducted. I have few comments :

1- Regarding the MIC determination : it is very unusual to perform MIC using large volumes. Normally, MIC is determined using micro-fold dilution in 96-well plates. Why did the authors used a large size of well ? Do the authors have a reference proving that they perform the assay according to a specific recommendation as EUCAST recommandation of MIC in 96-well plates. If not, the authors must redo the MIC assay using standardized procedure such as EUCAST one.

2- Table 4 : The numbers in Table 4 do not match the values given in the text and in the abstract. Please check.

3- In addition to test the antiviral action (by incubating the virus and the cells with nano), did the authors tested the viricidal activity of the nano by exposing the virus to nano before infecting the cells ?

Regards

Reviewer 3 Report

  1. This article has a lot of experimental methods, but the result data is very small. This is very strange.
  2. Most of the results are only the data of HARFHPßCD/AA@PLGA-PEG NPs. Is there any data for PLGA-PEG NPs? Both data should be provided to compare the differences.
  3. Why was HARF-HPßCD made into composite film first? In order to make HARF-HPßCD/AA@PLGA-PEG nanoparticles, the composite film was further dissolved. Isn't it simpler to mix HARF, HPßCD, and AA in water directly?
  4. A good study should not have only one condition. For example, you can compare the ratio of AA/HARF-HPßCD complex to analyze the antibacterial and antiviral abilities.

Round 2

Reviewer 2 Report

Dear Editor,

the authors addressed my concerns. I endorse the publication.

regards

Marc Maresca

Author Response

We thank the reviewer for this positive comment.

Reviewer 3 Report

The authors must compare more conditions.

Author Response

Thanks for your comment. However, this is beyond the scope of this paper and may be considered for future studies.